# Evaluation of the Interaction between Carvacrol and Thymol, Major Compounds of *Ptychotis verticillata* Essential Oil: Antioxidant, Anti-Inflammatory and Anticancer Activities against Breast Cancer Lines

**DOI:** 10.3390/life14081037

**Published:** 2024-08-20

**Authors:** Mohamed Taibi, Amine Elbouzidi, Mounir Haddou, Abdellah Baraich, Douaae Ou-Yahia, Reda Bellaouchi, Ramzi A. Mothana, Hanan M. Al-Yousef, Abdeslam Asehraou, Mohamed Addi, Bouchra El Guerrouj, Khalid Chaabane

**Affiliations:** 1Laboratoire d’Amélioration des Productions Agricoles, Biotechnologie et Environnement (LAPABE), Faculté des Sciences, Université Mohammed Premier, Oujda 60000, Morocco; amine.elbouzidi@ump.ac.ma (A.E.); haddou.mounir27@gmail.com (M.H.); elguerroujb@gmail.com (B.E.G.); k.chaabane@ump.ac.ma (K.C.); 2Centre de l’Oriental des Sciences et Technologies de l’Eau et de l’Environnement (COSTEE), Université Mohammed Premier, Oujda 60000, Morocco; 3Department of Biological Engineering, IUT Saint-Brieuc, University of Rennes, 35000 Rennes, France; abdellah.baraich@ump.ac.ma (A.B.); douaae.ou-yahia@univ-rennes1.fr (D.O.-Y.); asehraou@yahoo.fr (A.A.); 4Laboratory of Bioresources, Biotechnology, Ethnopharmacology and Health, Faculty of Sciences, Mohammed First University, Oujda 60000, Morocco; r.bellaouchi@ump.ac.ma; 5Department of Pharmacognosy, College of Pharmacy, King Saud University, Riyadh 11451, Saudi Arabia; rmothana@ksu.edu.sa (R.A.M.); halyousef@ksu.edu.sa (H.M.A.-Y.)

**Keywords:** antioxidant, anti-inflammatory, anticancer, thymol, carvacrol, mixture, cytotoxicity, selectivity, breast cancer, natural therapies

## Abstract

The objective of this study was to evaluate the antioxidant, anti-inflammatory, and anticancer properties of thymol, carvacrol, and their equimolar mixture. Antioxidant activities were assessed using the DPPH, ABTS, and ORAC methods. The thymol/carvacrol mixture exhibited significant synergism, surpassing the individual compounds and ascorbic acid in DPPH (IC_50_ = 43.82 ± 2.41 µg/mL) and ABTS (IC_50_ = 23.29 ± 0.71 µg/mL) assays. Anti-inflammatory activity was evaluated by inhibiting the 5-LOX, COX-1, and COX-2 enzymes. The equimolar mixture showed the strongest inhibition of 5-LOX (IC_50_ = 8.46 ± 0.92 µg/mL) and substantial inhibition of COX-1 (IC_50_ = 15.23 ± 2.34 µg/mL) and COX-2 (IC50 = 14.53 ± 2.42 µg/mL), indicating a synergistic effect. Anticancer activity was tested on MCF-7, MDA-MB-231, and MDA-MB-436 breast cancer cell lines using the MTT assay. The thymol/carvacrol mixture demonstrated superior cytotoxicity (IC_50_ = 0.92–1.70 µg/mL) and increased selectivity compared to cisplatin, with high selectivity indices (144.88–267.71). These results underscore the promising therapeutic potential of the thymol/carvacrol combination, particularly for its synergistic antioxidant, anti-inflammatory, and anticancer properties against breast cancer. This study paves the way for developing natural therapies against breast cancer and other conditions associated with oxidative stress and inflammation, leveraging the synergistic effects of natural compounds like thymol and carvacrol.

## 1. Introduction

Cancer represents one of the most formidable global public health challenges [1]. According to the World Health Organization, it is the second leading cause of death worldwide, with its incidence and prevalence steadily increasing [2]. The intricate and diverse mechanisms involved in carcinogenesis make it a highly active field of research, essential for the development of innovative therapeutic strategies [3]. Among the various types of cancer, breast cancer is particularly prevalent among women, accounting for approximately 25% of all newly diagnosed cancer cases each year [4]. It is crucial to recognize that breast cancer is not a single disease but comprises several distinct types, each with unique molecular, histological, and prognostic characteristics [5]. These types include ductal carcinomas in situ, lobular carcinomas in situ, invasive ductal carcinomas (or non-specific types), invasive lobular carcinomas, as well as less common forms such as tubular, medullary, or mucinous carcinomas [6]. This diversity highlights the need for tailored and personalized therapeutic approaches to effectively address the specific characteristics and behaviors of each breast cancer subtype.

Oxidative stress is a critical factor in the development of various illnesses, including cancer [7]. The condition is caused by an unequal production of reactive oxygen species (ROS) and the body’s capacity to counteract these molecules or fix the harm they inflict [8]. In the context of breast cancer, oxidative stress has a role in the beginning of tumor formation by causing DNA damage, facilitates the advancement of the disease by promoting uncontrolled cell growth, and supports the spread of cancer cells to other parts of the body by changing the surrounding environment of the tumor [9]. Furthermore, oxidative stress has a direct effect on the body’s ability to resist anticancer treatments, emphasizing its importance as a possible focus for therapeutic intervention [10].

Concurrently, inflammation is acknowledged as a significant risk factor in the progression of several diseases, such as cancer [11]. This issue is especially relevant in the context of breast cancer [12]. Chronic inflammation can contribute to the development of breast cancer through various pathways, including causing DNA damage, promoting the growth of new blood vessels (angiogenesis), and affecting the immune system [13]. The link between inflammation and breast cancer is reciprocal: inflammation can promote tumor formation, while cancer cells can, in turn, maintain a persistent state of inflammation [14].

Considering these difficulties, there is a growing inclination to explore aromatic and medicinal herbs as alternative therapeutic options [15]. These plants are well known for having many bioactive molecules, which provide a wide range of chemicals with diverse pharmacological effects [16]. Within this group of chemicals, secondary metabolites such as polyphenols, terpenes, and alkaloids have demonstrated encouraging effects in the prevention and treatment of several diseases, including cancer [17]. Carvacrol (2-methyl-5-(1-methylethyl)phenol) and thymol (2-isopropyl-5-methylphenol), two phenolic monoterpenes naturally present in the essential oils of various aromatic plants, such as Ptychotis verticillata, are of particular interest in this context [18,19]. These two compounds are positional isomers, sharing a similar chemical structure with a hydroxyl group and an isopropyl group attached to a benzene ring [20].

Carvacrol and thymol are known to have many interesting biological activities, including antimicrobial, antioxidant, anti-inflammatory, and analgesic properties. Carvacrol and thymol are known to have many interesting biological activities, including antimicrobial, antioxidant, anti-inflammatory and analgesic properties [21,22]. These properties make them promising compounds for numerous pharmaceutical and cosmetic applications [23]. This is why we have chosen to study the biological activities of carvacrol and thymol in more detail, both alone and in combination, in order to assess their potential use as natural therapeutic agents [24,25].

The use of an equimolar (1:1) blend of thymol and carvacrol is of particular interest as a natural option. This proportion enables their synergy to be exploited to the fullest, demonstrating exceptional results for antioxidant, anti-inflammatory and anticancer activities. The blend takes advantage of molecular interactions, significantly enhancing efficacy. This “whole-plant” approach preserves natural synergies and is often preferred [26]. Furthermore, the use of this natural blend offers the advantage of being a sustainable and ecological option, responding to the concerns of consumers and industries alike [27]. Thus, the development of products based on this blend has great potential as a natural alternative for the treatment of pathologies linked to oxidative stress, inflammation, and cancer.

The objective of this study is to comprehensively evaluate the antioxidant, anti-inflammatory, and anticancer activities of carvacrol and thymol, the primary compounds found in the essential oil of *Ptychotis verticillata* Duby, along with their equimolar mixture (50%/50%), on breast cancer cell lines. The objective of this investigation is to analyze the antioxidant properties of these compounds and their combination using laboratory methods, evaluate their ability to regulate inflammatory responses in appropriate cellular models, and determine their efficacy against different breast cancer cell lines in terms of their anticancer activity.

## 2. Materials and Methods

### 2.1. Chemicals

All chemicals used in this study were of analytical grade, unless otherwise specified, and were used as is, without further purification. Carvacrol (purity: 99%) and thymol (purity: 98.5%) were purchased from Sigma-Aldrich (St. Louis, MO, USA). Sigma-Aldrich is renowned for supplying high-quality reagents and chemicals for scientific research. The use of these high-purity reagents is essential to guarantee the integrity of experimental results and ensure that the effects observed are directly attributable to the compounds studied. Compliance with strict compound purity standards ensures the reliability and reproducibility of the data obtained.

### 2.2. Physicochemical, Drug-Likeness, Pharmacokinetics, and In Silico Toxicity

Physicochemical characteristics of a molecule dictate its drug-likeness, its pharmacokinetics, and its potential toxicity. ADME (absorption, distribution, metabolism, and excretion) are key elements in determining the pharmacokinetic features of a chemical, encompassing its processes of entering, circulating, metabolizing, and exiting the body. Computational approaches are applied to anticipate these features, including a compound’s capacity to permeate cell membranes, interact with transporters and enzymes involved in drug absorption and disposal, and maintain metabolic stability. To examine these drugs, we applied ADME prediction methods such as SwissADME and pkCSM [28,29]. These platforms permitted the investigation of physicochemical features, drug similarity, and pharmacokinetic behavior. For analyzing toxicity levels, we applied the Pro-Tox II web tool (accessed on 21 March 2023) [30]. This program applies statistical algorithms to compare a substance’s chemical structure with a comprehensive database of known hazardous substances, predicting the chance of generating toxicity or bad effects in humans or other creatures. It contains information on LD_50_ values, toxicity classes, and specific toxicological endpoints such as hepatotoxicity, nephrotoxicity, carcinogenicity, immunotoxicity, mutagenicity, and cytotoxicity.

### 2.3. Antioxidant Activity Protocols

#### 2.3.1. DPPH Scavenging Capacity

The method utilized to test the compound’s efficacy in scavenging DPPH radicals was based on protocols detailed in references [31,32,33,34], with minor changes. The test was completed in duplicate for each concentration. A solution of 0.1 mM DPPH (2,2-diphenyl-1-picrylhydrazyl) was produced, and the compounds were evaluated at different concentrations ranging from 1 to 500 µg/mL. Ascorbic acid served as the positive control with concentrations ranging from 0.01 to 300 µg/mL [35].

#### 2.3.2. ABTS Scavenging Activity

With a few minor adjustments [36], the procedure published by Re et al. [37], was used to evaluate the test samples’ capacity to scavenge radicals against the ABTS (2,2′-azinobis-3-ethylbenzothiazoline-6-sulphonate) radical cation. IC_50_ (mg/L) was the unit of measurement for the radical scavenging capacity, with ascorbic acid acting as reference standard with concentrations ranging from 0.01 to 300 µg/mL.

#### 2.3.3. Oxygen Radical Absorbance Capacity

The ORAC (Oxygen Radical Absorbance Capacity) test was carried out using a modified version of the procedure reported by Huang et al. [38]. This test examines the antioxidant capacity of compounds by examining their ability to neutralize oxygen radicals. The method was carried out utilizing a 96-well microplate format. First, 150 µL of fluorescein solution, produced as 8 × 10^−8^ M fluorescein in 0.075 M phosphate buffer (pH 7.4), were dispersed to each well. Fluorescein is a fluorescent probe whose fluorescence lowers when it reacts with radicals. Next, 25 µL of the substance or mixture to be evaluated was added to the corresponding wells. Wells containing 25 µL of phosphate buffer served as blanks, and wells containing 25 µL of Trolox solution (from 10 to 100 µm) were utilized as standards. Control wells containing 50 µL phosphate buffer without any radicals were also added to provide a baseline. The reaction was initiated by adding 25 µL of a freshly produced solution of AAPH (2,2′-azobis(2-methylpropionamidine) dihydrochloride) at a concentration of 165.94 mM in phosphate buffer to each well, except for the control wells, at a temperature of 37 °C. AAPH creates peroxyl radicals that induce oxidative stress in the fluorescein solution, simulating free-radical attack. After addition of AAPH, the microplate was shaken for 8 s to ensure full mixing of the reagents. The fluorescence in-tensity of the mixture was then monitored at regular intervals for 120 min.

### 2.4. Anti-Inflammatory Activity

#### 2.4.1. 5-Lipoxygenase (5-LOX) Inhibition

The study of the anti-inflammatory activity of carvacrol, thymol, and their equimolar mixture was carried out by focusing on the inhibition of the 5-LOX enzyme. The method is based on the protocol described by Pinto et al. [39], with slight modifications to suit our specific demands. Enzyme activity was evaluated using a UV–visible spectrophotometer (Jenway model 6300) set at 560 nm, following the production of the Fe^3+^/xylenol orange combination. The experimental technique contained three critical steps: firstly, the enzyme 15-lipoxygenase isolated from soybean (Glycine max) was incubated with carvacrol, thymol, their equimolar combination, or a conventional inhibitor for 5 min at 25 °C. Next, linoleic acid was added to a final concentration of 140 μM in 50 mM Tris-HCl buffer (pH 7.4), followed by a 20 min incubation at 25 °C in the dark. The reaction was halted by the addition of 100 μL of FOX reagent, consisting of a methanol/water combination (9:1 *v/v*) containing 30 mM sulfuric acid, 100 μM xylenol orange, and 100 μM iron (II) sulfate. Control wells containing simply the buffer and LOX solution, as well as blanks where the substrate was introduced after the FOX reagent, were included in each series of experiments to establish the reliability of the data. After a final 30 min incubation at 25 °C, absorbance at 560 nm was measured [40].

The % inhibition of hydrogen peroxide formation, an indication of lipoxygenase inhibitory action, was computed according to the formula:$$Inhibition\;percentage\left(\%\right)=\left[\frac{\left(A_{control}-A_{blank}\right)-\left(A_{sample}-A_{blank}\right)}{\left(A_{control}-A_{blank}\right)}\right]$$
where A_control_, A_blank_ and A_Sample_ represent the absorbances of the control well, blank, and test sample, respectively. Quercetin was used as a reference at concentrations from 3.125 to 300 µg/mL.

#### 2.4.2. Cyclooxygenase 1/2 (COX-1/2) Inhibition

The COX inhibitory effects of the substances were examined using a COX-1/COX-2-catalyzed prostaglandin production test, using a modified approach by Futaki et al. [41]. The chemicals were dissolved in 4% DMSO to a final concentration of 100 µg/mL.

In this test, one unit of COX-1 or COX-2 enzyme was suspended in a 0.1 M Tris-HCl buffer (pH 7.5) containing hematin (1 mM) and phenol (2 mM) as cofactors. The COX-1 enzyme was produced from ram seminal vesicles, while COX-2 was taken from sheep placenta with a purity of 70% (both from Cayman Chemical Company, Ann Arbor, MI, USA). The reaction mixture was preincubated with either the solvent vehicle (4% DMSO) or the test solution for 2 min at 37 °C. Following preincubation, 51.4 µM [1–14C] arachidonic acid (Sigma, St. Louis, MO, USA) was added to activate the reaction. Arachidonic acid, tagged with radioactive carbon ([1–14C]), serves as a substrate for the COX enzymes. The mixture was then incubated for an additional 2 min at 37 °C. The process was terminated by adding 400 µL of an n-hexane and ethyl acetate combination (2:1, *v/v*) to extract prostaglandin E2 (PGE2). This was followed by centrifugation at 2000 rpm for 1 min to separate the phases. The organic solvent phase was removed, and the PGE_2_ level in the remaining aqueous phase was quantified using a radioimmunoassay with a liquid scintillation counter, which measures the radioactivity (disintegrations per minute, DPM) of the [1–14C]-labeled PGE2. The percentage inhibition of COX-catalyzed prostaglandin production by the test compounds was estimated by comparing the radioactivity of the samples with the solvent vehicle controls. A decrease in radioactivity shows successful COX enzyme inhibition by the test substances. Indomethacin, a recognized non-steroidal anti-inflammatory medication (NSAID), was employed as a positive control to verify the assay. The later compound was used at concentration varying from 3.90 to 500 µg/mL. The inhibitory potency of the test substances was evaluated by measuring their IC_50_ values, which represent the concentration needed to inhibit 50% of enzyme activity. IC_50_ values were acquired using dose–response curve analysis, providing a measure of the drugs’ efficiency as COX inhibitors.

### 2.5. Anticancer Activity

#### 2.5.1. Cell Lines and Culture Conditions

Human breast cancer cell lines, including MCF-7 (ER+), MDA-MB-231 (triple negative), and MDA-MB-468, were obtained from the American Type Culture Collection (ATCC, Manassas, VA, USA). These cell lines were selected to represent distinct sub-types of breast cancer for extensive investigation. The cells were grown in RPMI 1640 medium, acquired from Gibco, Thermo Fisher Scientific (Waltham, MA, USA). The medium was added with 10% heat-inactivated fetal calf serum (FCS) to supply vital growth hormones and nutrients necessary for cell proliferation. Additionally, 1% penicillin streptomycin was administered to avoid bacterial contamination, ensuring the sterility and integrity of the cell cultures. The cell cultures were maintained at 37 °C in a humidified environment containing 5% CO_2_. This controlled environment mimics physiological conditions in the human body, supporting optimal cell development and function. The humidified atmosphere prevents medium evaporation, while the 5% CO_2_ concentration maintains optimal pH values in the culture medium. These specialized circumstances and methods ensure that the breast cancer cell lines remain healthy and viable for experimental procedures, delivering dependable and reproducible results for research projects. The standard protocols followed for cultivating these cells are crucial for generating consistent and reliable data. Using well-characterized cell lines from a credible source like ATCC ensures the re-liability and comparability of the experimental outcomes [42,43].

#### 2.5.2. Evaluation of Cell Viability

The anticancer activity of carvacrol, thymol, and its equimolar mixture was assessed by the MTT (3-(4,5-dimethylthiazol-2-yl)-2,5-diphenyltetrazolium bromide) experiment using a method derived from Mosmann (1983) [44]. This approach determines cell viability based on the ability of living cells to decrease MTT to a purple, water-insoluble formazan product. Cells were planted in 96-well plates at a density of 1 × 10^4^ cells per well in 100 μL of complete culture media, which provides all necessary nutrients for cell growth. After a 24 h incubation period to allow for cell adhesion, the medium was replaced with fresh medium containing various concentrations of carvacrol, thymol, or their equimolar mixture (6.25, 12.5, 25, 50, 100, and 200 μg/mL), all previously diluted in DMSO to achieve a final DMSO concentration of ≤0.1%. The cells were then treated with the compounds for 24, 48, and 72 h to determine the effect on cell viability over varied time intervals. Following each incubation time, 20 μL of MTT solution (at a concentration of 5 mg/mL in PBS) were applied to each well. The plates were further incubated for 4 h at 37 °C, during which time live cells decrease the MTT to formazan, creating purple crystals. At the end of the incubation, the medium containing the MTT was carefully aspirated, leaving only the formazan crystals in the bottom of the wells. To dissolve these crystals, 100 μL of DMSO were administered to each well. The resulting color intensity, which is proportional to the amount of formazan produced and hence to the number of live cells, was quantified at a wavelength of 570 nm using a microplate reader (Synergy HT, BioTek Instruments, Winooski, VT, USA). Cell viability was determined using the following formula:$$Cell\;viability\left(\%\right)=100-\left[\left(\frac{A_{0}-A_{t}}{A_{0}}\right)\times 100\right]$$
where A_t_ where is the absorbance of cells treated with the test compounds at various doses and A_0_ is the absorbance of cells treated with 0.1% DMSO. This formula quantifies the reduction in cell viability in response to the treatments compared to the control. By comparing these values, the relative anticancer efficacy of carvacrol, thymol, and their equimolar mixture can be determined. Cisplatin, a known anticancer drug, was used as a reference drug at varying concentrations from 0.01 to 50 µg/mL.

## 3. Results and Discussion

### 3.1. Physico-Chemical Properties and Drug-Likeness of Thymol and Carvacrol

In silico drug-likeness assessments serve a significant role in enhancing drug discovery by quickly selecting and prioritizing promising candidates. These computational tools expedite operations, saving time and money, lowering experimental workload, and enabling the early discovery of compounds with favorable pharmacokinetic features and target interactions. By exploiting these tools, researchers can fast filter down enormous compound libraries to a select number of potential candidates, hence accelerating the medication development process.

Lipinski’s rule of five outlines specific physical and chemical characteristics essential for oral bioavailability in humans: fewer than five H-bond donors, fewer than 10 H-bond acceptors, no more than 10 nitrogen or oxygen atoms, a molecular weight under 500 Daltons, and a calculated partition coefficient (MLOGP) of 4.15 or less. Adherence to these criteria predicts the likelihood that a chemical would exhibit oral action. Importantly, all phytoconstituents described in this study adhere to Lipinski’s rule of five, as detailed in Table 1. This adherence highlights that these substances contain the requisite properties for optimal oral bioavailability. In addition to Lipinski’s rule, the compounds were analyzed against other drug-likeness criteria such as Veber’s and Egan’s guidelines. Veber’s rule highlights the number of rotatable bonds and the topological polar surface area (TPSA), stating that compounds with 10 or fewer rotatable bonds and a TPSA of 140 Å² or less are likely to display good oral bioavailability. Egan’s rule, which analyzes solubility and permeability, is another key metric. All compounds found in this investigation adhere to these additional requirements, further substantiating their potential as viable medication candidates.

According to Martin’s 2005 research, every chemical matching Lipinski’s rule of five is assigned a bioavailability score of 0.55. This score acts as a trustworthy measure of a compound’s capability for oral absorption. As a result, all substances found in this investigation obtain a bioavailability score of 0.55. Table 1. displays the bioavailability radars for these drugs, where the pink zone represents the ideal space for oral bioavailability. For a molecule to be called drug-like, its radar plot must totally lie within this area. This visual representation gives a rapid and comprehensive technique for evaluating the drug-likeness of numerous substances.

The table gives a full comparative investigation of the physicochemical characteristics, drug-likeness, and bioavailability of thymol and carvacrol. These two compounds share a similar chemical structure, varying only in the location of the hydroxyl group. The structural representations of thymol and carvacrol illustrate this distinction while illustrating their overall resemblance. The physicochemical characteristics of thymol and carvacrol are nearly similar. Both compounds have a molecular weight of 150.221 g/mol and a TPSA of 20.23 Å². Each possesses one hydrogen bond acceptor and one hydrogen bond donor. Additionally, both compounds have one rotatable bond and a LogP value of 2.82, showing their lipophilicity. These features suggest that thymol and carvacrol exhibit comparable behavior in biological systems, making them viable candidates for further drug development [45,46].

The drug-likeness of thymol and carvacrol was analyzed using numerous criteria, including Lipinski’s, Egan’s, and Veber’s rules [47]. Both compounds conform with these requirements, indicating good qualities for possible oral medication candidates. Lipinski’s rule measures drug-likeness based on molecular weight, lipophilicity, hydrogen bond donors, and acceptors. Egan’s rule examines bioavailability, focusing on solubility and permeability, while Veber’s rule predicts oral bioavailability based on the amount of rotatable bonds and TPSA. The compliance of thymol and carvacrol with these principles shows they contain features conducive to good oral bioavailability, making them strong candidates for further development.

The bioavailability score for both thymol and carvacrol is 0.55, indicating moderate bioavailability [48]. This score, paired with the bioavailability radar maps, suggests that these compounds have a balanced profile across numerous critical criteria, including lipophilicity, polarity, insolubility, size, flexibility, and unsaturation. The radar charts vividly depict the compliance of thymol and carvacrol with numerous drug-likeness and bioavailability characteristics, further demonstrating their potential as drug candidates. This study emphasizes that both thymol and carvacrol comply to the required requirements for oral bioavailability, suggesting their potential applicability as therapeutic candidates. Additionally, the similarity in their physicochemical features and compliance with various drug-likeness requirements supports the argument for their further exploration and development as possible therapeutic agents.

### 3.2. Pharmacokinetics (ADME Properties) of Thymol and Carvacrol and Their Potential Toxicity

Assessing the pharmacokinetic characteristics of small molecules is crucial in drug development and high-throughput screening. Pharmacokinetics encompass the absorption, distribution, metabolism, and excretion (ADME) of drugs in the body, closely linked with their toxicity profile. Predicting ADME and toxicity properties using computational methods has emerged as a valuable approach due to the time and cost associated with obtaining such data from in vitro, in vivo, or preclinical studies.

Regarding the pharmacokinetic properties of both thymol and carvacrol, the compounds exhibit different water solubility values, with thymol having a log mol/L solubility of 3.01 and carvacrol 3.31, indicating moderate solubility for both compounds (Table 2). The solubility class for both compounds is classified as moderately soluble (M.S.). The Caco-2 permeability values are 4.44 for thymol and 4.41 for carvacrol, suggesting that both compounds have high intestinal permeability. Additionally, both thymol and carvacrol are predicted to be absorbed in the intestine, which is consistent with their high permeability values. The log Kp (cm/s) values for thymol and carvacrol are −2.64 and −2.64, respectively, indicating similar skin permeability for both compounds.

The distribution-related parameters show that the volume of distribution (VDss) is 2.91 L/kg for thymol and 2.72 L/kg for carvacrol, suggesting that both compounds distribute moderately within the body. The blood–brain barrier (BBB) permeability values are 1.82 for thymol and 1.99 for carvacrol, indicating that both compounds have moderate permeability across the BBB. In terms of metabolism-related parameters, neither thymol nor carvacrol are substrates or inhibitors of CYP2D6 and CYP3A4 enzymes, suggesting minimal interaction with these major metabolic enzymes.

The excretion-related parameters reveal that the total clearance values are 8.67 mL/min/kg for thymol and 8.65 mL/min/kg for carvacrol, indicating similar clearance rates for both compounds. Both thymol and carvacrol are non-inhibitors of the renal OCT2 substrate, suggesting that renal excretion of these compounds is unlikely to be affected by OCT2 inhibition.

The toxicity-related parameters indicate that the predicted LD50 values are 640 mg/kg for thymol and 810 mg/kg for carvacrol, suggesting that carvacrol may be slightly less toxic than thymol. Both compounds have a predicted toxicity class of 4, indicating moderate toxicity. The probability values for various toxic effects such as hepatotoxicity, carcinogenicity, immunotoxicity, mutagenicity, and cytotoxicity are relatively low, suggesting that thymol and carvacrol have a low likelihood of causing these adverse effects. Specifically, the probability of hepatotoxicity is 0.75 for both compounds, while the probability of carcinogenicity is 0.02, indicating a low risk of cancer-causing potential. Both thymol and carvacrol are predicted to be inactive in terms of immunotoxicity and mutagenicity, with probabilities of 0.60 and 0.99, respectively. The cytotoxicity probabilities are also low at 0.98 for thymol and 0.89 for carvacrol, suggesting minimal cytotoxic effects.

In summary, the in silico pharmacokinetic predictions indicate that thymol and carvacrol have favorable absorption and distribution properties, with moderate water solubility and high intestinal permeability. Both compounds have similar clearance rates and are non-inhibitors of the renal OCT2 substrate. The predicted toxicity profiles suggest that thymol and carvacrol have low probabilities of causing hepatotoxicity, carcinogenicity, immunotoxicity, mutagenicity, and cytotoxicity. Overall, these findings support the potential use of thymol and carvacrol as therapeutic agents, although further in vivo studies are necessary to confirm these predictions.

### 3.3. Antioxidant Activity Results

Plant-derived phenolic compounds thymol and carvacrol have garnered significant interest for their potential to combat oxidative stress, a process associated with numerous chronic illnesses. This research assessed the antioxidant capabilities of thymol, carvacrol, and an equimolar mixture of both using three well-established methods: DPPH and ABTS radical scavenging tests, as well as the oxygen radical absorbance capacity (ORAC) assay. These experiments aimed to elucidate how these substances might help protect against harmful oxidative processes in the body. For DPPH radical scavenging capacity, the equimolar mixture of thymol/carvacrol (1:1) exhibited higher activity (IC_50_ = 43.82 ± 2.41 µg/mL) compared to the individual compounds and ascorbic acid used as a positive control. Thymol alone (IC_50_ = 161.02 ± 6.89 µg/mL) demonstrated better activity than carvacrol (IC_50_ = 249.09 ± 9.04 µg/mL) and ascorbic acid (IC_50_ = 188.24 ± 6.46 µg/mL) (Table 3). A similar trend was observed in the ABTS radical scavenging activity, where the thymol/carvacrol mixture showed remarkable efficacy (IC_50_ = 23.29 ± 0.71 µg/mL), slightly surpassing ascorbic acid (IC_50_ = 25.20 ± 3.06 µg/mL). In this test, carvacrol alone (IC_50_ = 107.88 ± 4.46 µg/mL) proved more effective than thymol (IC_50_ = 125.31 ± 6.25 µg/mL). Concerning the ORAC assay, the results revealed that carvacrol possesses the highest activity (3.535 ± 0.127 mmol TE/g), followed by thymol (2.341 ± 0.009 mmol TE/g). Interestingly, the thymol/carvacrol mixture showed lower ORAC activity (0.273 ± 0.096 mmol TE/g) than the individual compounds.

These results show that both carvacrol and thymol possess intrinsic antioxidant properties. Carvacrol can directly neutralize free radicals thanks to its phenolic hydroxyl group and chelate transition metals, thus limiting radical formation [49]. Thymol shares this free radical scavenging capacity, but can also activate endogenous enzymatic antioxidant defense systems [50].

When carvacrol and thymol act synergistically, their mechanisms of action complement each other to offer enhanced antioxidant protection [51]. Their similar chemical structure enables them to work synergistically to neutralize free radicals more effectively [52]. Furthermore, the molecular interactions between these two compounds mutually stabilize their respective antioxidant properties [53].

These results confirm other recent studies on the antioxidant activity of thymol and carvacrol. For example, the study by Yildiz et al. (2021) also demonstrated the notable activity of these two compounds, showing antioxidant activities similar to those of BHT and BHA in the linoleic acid emulsion test at different concentrations [54]. Similarly, the results presented by Al-Mansori et al. (2020) also highlighted that carvacrol and thymol possess significant antioxidant activity and synergistic effects, confirmed by three different tests, namely, DPPH, FRAP, and TEAC [55].

Our results suggest a synergistic effect between thymol and carvacrol in the DPPH and ABTS assays, where their combination significantly enhanced antioxidant activity compared with the individual compounds. This synergistic effect could be attributed to the complementary mechanisms of action of the two molecules, or to their ability to mutually regenerate their active forms [56]. However, the antagonistic effect observed in the ORAC test for the thymol/carvacrol mixture is unexpected and merits further investigation. It could be explained by complex interactions between the two compounds under specific oxidizing conditions, or by limitations of the ORAC method for this type of mixture [57,58].

The best synergy between thymol and carvacrol appears to be in their free radical scavenging capacity. As shown in Table 1, the DPPH radical scavenging potential is significantly higher for their equimolar (1:1) combination compared to their individual use. This remarkable synergy in antioxidant activity is not, however, equally reflected in the results for anti-inflammatory activity.

Although synergy is also observed in inflammatory inhibition tests, it appears to be of lesser magnitude compared to antioxidant activity. This difference in behavior suggests that the underlying mechanisms involved in these two types of biological activity do not benefit from the same synergy when thymol and carvacrol are used in combination.

Importantly, this notable antioxidant activity could have significant therapeutic implications, particularly in the fight against breast cancer. Indeed, oxidative stress plays a crucial role in carcinogenesis and tumor progression [59]. Potent antioxidants such as thymol and carvacrol could potentially help reduce this oxidative stress, inhibit cancer cell proliferation and promote apoptosis. 

### 3.4. Anti-Inflammatory Activity Results

The purpose of this study was to examine the anti-inflammatory action of carvacrol, thymol, and its equimolar mixture (50% thymol/50% carvacrol) on three important pro-inflammatory enzymes: 5-lipoxygenase (5-LOX), cyclooxygenase-1 (COX-1), and cy-clooxygenase-2 (COX-2). Quercetin and indomethacin were used as positive controls (Figure 1). The results demonstrate that the equimolar mixture exhibits the highest 5-LOX inhibition with an IC_50_ of 8.46 ± 0.92 µg/mL, surpassing quercetin, which has an IC_50_ of 10.17 ± 0.78 µg/mL. Thymol (IC_50_ = 46.32 ± 1.46 µg/mL) and carvacrol (IC_50_ = 55.75 ± 2.72 µg/mL) both displayed considerable inhibition, which might be due to their comparable chemical structure and probable interaction with the 5-LOX enzyme. Regarding COX-1 inhibition, thymol and the equimolar mixture demonstrated considerable inhibition, with IC_50_ values of 29.74 ± 1.09 µg/mL and 15.23 ± 2.34 µg/mL, respectively. Carvacrol, with an IC_50_ of 70.53 ± 0.75 µg/mL, demonstrated lesser inhibition, presumably indicating a less efficient interaction with COX-1. For COX-2 inhibition, thymol (IC_50_ = 26.54 ± 3.21 µg/mL), carvacrol (IC_50_ = 24.63 ± 2.56 µg/mL), and the equimolar mixture (IC_50_ = 14.53 ± 2.42 µg/mL) displayed comparable inhibition, demonstrating a considerable ability to inhibit COX-2. These results imply that these substances may have various anti-inflammatory pathways by inhibiting both COX-1 and COX-2, albeit less efficiently than indomethacin. Notably, while carvacrol and thymol showed considerable outcomes, their mixture proved more effective in suppressing COX-2.

The results for thymol and carvacrol indicate notable inhibition of all three enzymes studied, suggesting that they could be useful in applications where partial inhibition of these pathways is desirable, perhaps in combination with other agents for therapeutic synergy. Furthermore, the data show that the equimolar mixture of thymol and carvacrol is a particularly potent inhibitor of 5-LOX, which could have significant implications for the development of new anti-inflammatory agents targeting this specific pathway. The synergistic effect observed with the thymol/carvacrol mixture, particularly pronounced for 5-LOX and COX-2 inhibition, is an important finding of this study. This phenomenon, confirmed by the significantly lower IC_50_ values of the mixture compared with the individual compounds, could be explained by complementary mechanisms of action, or by optimized interaction with the active enzyme sites in the presence of both compounds (Figure 2).

The probable mechanism of action for the significant anti-inflammatory activity of these two compounds, lies primarily in the inhibition of key enzymes of the arachidonic acid pathway: 5-lipoxygenase (5-LOX), cyclooxygenase-1 (COX-1), and cyclooxygenase-2 (COX-2) [60,61,62]. By inhibiting 5-LOX, these compounds reduce the production of leukotrienes, pro-inflammatory mediators associated with pathologies such as asthma and allergies [63]. In addition, although they may slightly inhibit COX-1, which could limit side effects on gastric mucosa and other physiological functions, their action is mainly directed against COX-2, an inducible enzyme that catalyzes the production of pro-inflammatory prostaglandins [64,65]. By specifically targeting COX-2, thymol and carvacrol can effectively reduce pain and inflammation while preserving the protective functions of COX-1 [66]. The equimolar (1:1) mixture of these two compounds seems to enhance their ability to inhibit the 5-LOX, COX-1, and COX-2 enzymes, enabling a more significant reduction in inflammation [67]. This synergy maximizes the anti-inflammatory effect while minimizing side effects, notably by preserving COX-1 activity [68].

These in vitro results are validated by multiple in vivo investigations indicating the anti-inflammatory activity of carvacrol. For example, the study by da Silva et al. (2012) indicated that carvacrol can act on several pharmacological targets, possibly by interfering with the release and/or synthesis of inflammatory mediators such as prostanoids [62]. Furthermore, a recent study by Gunes-Bayir et al. [51], demonstrated significant anti-inflammatory effects generated by carvacrol at doses of 10 and 25 mg/kg body weight. These results not only interfered with N-methyl-N’-nitro-N-nitrosoguanidine (MNNG)-induced gastric carcinogenesis in Wistar rats but also offered hepatoprotection [51].

In the case of thymol, their remarkable results on anti-inflammatory activity corroborate other observations obtained by various evaluation methods. Lanzarin et al. (2023) demonstrated the anti-inflammatory efficacy of thymol on a transgenic line of zebrafish larvae (Tg(mpx)i114). In this study, thymol reduced the number of neutrophils, an inflammatory marker, at the wound site after caudal fin amputation [69].

Several studies have confirmed the synergy between carvacrol and thymol in their evaluation of various biological activities, corroborating the results obtained in our study [70,71]. Indeed, our research has shown that the combination of these two compounds results in significant inhibition of the enzymes studied, notably 5-lipoxygenase (5-LOX). This synergy could potentially be exploited to develop new therapeutic strategies, particularly in the field of anti-inflammation, where partial but targeted inhibition of enzymatic pathways is desirable. The results of our study are therefore in line with the literature data, reinforcing the idea that carvacrol and thymol, used in combination, have considerable potential for innovative medical applications.

This study provides in vitro evidence of the anti-inflammatory potential of carvacrol, thymol and their equimolar mixture, mainly via the inhibition of 5-LOX and COX-2. The thymol/carvacrol mixture appears particularly promising, demonstrating a significant synergistic effect with IC_50_ values lower than those of the individual compounds for all enzymes tested. These results, combined with in vivo observations reported in the literature, suggest that the combined use of these compounds could offer therapeutic benefits in modulating the inflammatory response.

### 3.5. Anticancer Activity against Breast Cancer Lines

The anticancer activity of thymol, carvacrol, and their equimolar combination was evaluated against various breast cancer cell lines, suggesting good therapeutic potential (Figure 2). The cell lines employed included MCF-7 (estrogen receptor-positive breast cancer), MDA-MB-231, and MDA-MB-436 (triple-negative breast tumors), as well as peripheral blood mononuclear cells (PBMC) as a normal cell control. Cisplatin, a well-established chemotherapeutic drug, was employed as a positive control.

The cytotoxicity of the substances was assessed by MTT assay, exhibiting variable effectiveness depending on the cell lines examined (Figure 3 and Table 4). IC50 values observed for the thymol/carvacrol combination were 0.92 ± 0.09 μg/mL for MCF-7 cells, 1.46 ± 0.16 μg/mL for MDA-MB-231 cells, and 1.70 ± 0.22 μg/mL for MDA-MB-436 cells. These data reveal a very high cytotoxic activity against all the breast cancer lines examined, surpassing that of cisplatin (IC_50_ = 8.43 ± 0.92 μg/mL for MCF-7) (Table 4).

The selectivity of the compounds towards cancer cells versus normal cells was assessed by calculating the selectivity index (SI) (Table 2), comparing the IC_50_ of PBMC with that of cancer lines. The SIs obtained for the thymol/carvacrol combination were remarkably high: 267.71 for MCF-7, 168.96 for MDA-MB-231, and 144.88 for MDA-MB-436. These values are significantly higher than those for cisplatin (3.01, 5.51, and 6.40 respectively), suggesting a greater selectivity of the thymol/carvacrol combination towards cancer cells.

A particularly interesting aspect of this study is the synergistic effect observed between thymol and carvacrol. The IC_50_ values of the equimolar combination (0.92 and 1.70 μg/mL) are significantly lower than those of the individual compounds (thymol: 5.16, and 15.77 μg/mL; carvacrol: 5.93, and 9.76 μg/mL), indicating a potentiation of the cytotoxic effect. This synergy could result from complementary mechanisms of action or mutual amplification of the effects of each compound. For instance, thymol and carvacrol could act on different cell signaling pathways, or mutually reinforce each other’s ability to disrupt cell membranes. This synergistic interaction offers promising prospects for the development of combination therapies that are more effective and potentially less toxic to healthy tissues.

The exceptional selectivity of the thymol/carvacrol combination, particularly towards MCF-7 cells, suggests that this combination could potentially target cancer cells while largely sparing normal cells, a highly desirable attribute for any potential anticancer agent. What is more, the combination’s superior efficacy versus cisplatin against all the breast cancer lines tested, combined with its enhanced selectivity, indicates a promising therapeutic potential, particularly in the treatment of breast cancer.

Carvacrol and thymol demonstrate complementary individual anticancer properties. On the one hand, carvacrol is able to induce apoptosis in tumor cells by activating apoptotic signaling pathways, such as those involving caspases [72]. It can also inhibit cell proliferation by impeding cell cycle progression [73]. On the other hand, thymol has the ability to reduce cancer cell migration and invasion by modulating the expression and activity of enzymes involved in extracellular matrix degradation, notably matrix metalloproteinases (MMPs) [74].

When carvacrol and thymol are studied in combination, their anticancer activity is remarkably synergistic. Their similar chemical structures enable them to act complementarily on different signaling pathways involved in carcinogenesis, such as those regulating apoptosis, proliferation and cell migration/invasion [75,76]. Moreover, the molecular interactions between these two compounds amplify their ability to induce apoptosis, inhibit proliferation and reduce tumor cell dissemination [77]. This synergy of action results in enhanced anticancer activity, suggesting a promising therapeutic potential against various subtypes of breast cancer.

These results corroborate data from prior research demonstrating the anticancer activity of thymol and carvacrol against various forms of cancer. Notably, the study by Mari et al. (2021) demonstrated that carvacrol dramatically reduced the survival of MCF-7 cells with an IC_50_ of 200 µmol/L at 24 and 48 h, triggering cell cycle arrest in the G0/G1 phase and inhibiting PI3K/p-AKT proteins. This led to a decrease in Bcl-2 expression and an increase in Bax, causing apoptosis via the PI3K/p-AKT pathway [78]. Additionally, the study by Seresht et al. (2019) verified thymol’s anticancer activity, reporting IC_50_ values of 54 and 62 μg/mL on MCF-7 cells for 48 h and 72 h, respectively [79]. Most investigations on these two drugs have focused on the MCF-7 cell line utilizing various methodologies. Investigations into the synergy between carvacrol and thymol have mostly explored different biological functions. However, our study is the first to evaluate the effects of an equimolar mixture of these chemicals on this breast cancer cell line. These results confirm that thymol and carvacrol can be used in conjunction as a natural option for the treatment of different forms of breast cancer.

## 4. Conclusions

The results the findings of this investigation underline the great therapeutic potential of an equimolar blend of thymol and carvacrol, the major ingredients of *Ptychotis vertic-illata* essential oil, particularly for its synergistic antioxidant, anti-inflammatory, and anticancer effects. Both thymol and carvacrol were shown in our in silico analysis to meet drug-likeness criteria (Lipinski’s rule of five, Veber’s rules, and Egan’s rule) and possess favorable pharmacokinetic profiles with moderate toxicity and a low risk of being carcinogenic, mutagenic, hepatotoxic, nephrotoxic, or immunotoxic. The mixture displayed excellent antioxidant activity, surpassing the individual components and ascorbic acid in both DPPH and ABTS experiments. Additionally, it displayed substantial anti-inflammatory effects by dramatically reducing the 5-LOX, COX-1, and COX-2 enzymes. Notably, the mix displayed exceptional anticancer efficacy against MCF-7, MDA-MB-231, and MDA-MB-436 breast cancer cell lines, demonstrating greater cytotoxicity and selectivity compared to cisplatin. This discovery presents new opportunities for the food, cosmetic, and pharmaceutical industries to develop innovative natural products that harness the synergistic effects of these bioactive compounds.

Future research should focus on further elucidating the precise molecular mechanisms underlying the observed synergies, as well as evaluating the efficacy and safety of the thymol/carvacrol combination in relevant animal models and clinical trials. Exploring additional therapeutic applications, such as their potential use in neurodegenerative, cardiovascular, or metabolic disorders, could also expand the scope of this promising natural remedy.

## Figures and Tables

**Figure 1 life-14-01037-f001:**
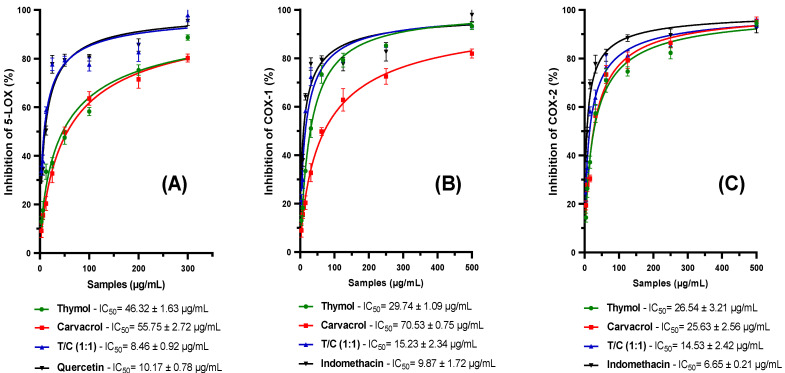
Results of the anti-inflammatory activity via COX/LOX pathway of both thymol and carvacrol along with their equimolar mixture (1:1), against 5-LOX (**A**), COX-1 (**B**), and COX-2 (**C**). Quercetin was used as a positive control for 5-LOX, while indomethacin was used as a positive control for COX-1/2. Data are presented as mean ± standard deviation. The experiments were conducted in three replicas.

**Figure 2 life-14-01037-f002:**
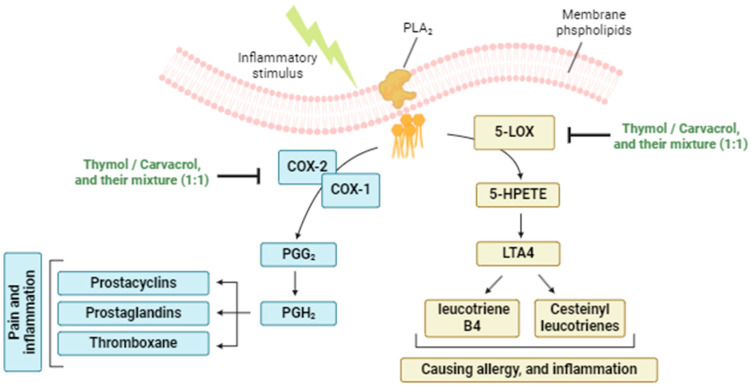
Simplified scheme of cyclooxygenase (COX) and lipoxygenase (LOX) pathways that are activated in response to inflammation, producing prostanoids and leukotrienes, respectively. PLA_2_: phospholipase A2; 5-LOX: 5-lipoxygenase; 5-HPETE: 5-hydroperoxyeicosatetraenoic acid; LTA4: leucotiene A4; COX-1/2: cyclooxygenase 1, and 2; PGG_2_: prostaglandin G_2_; PGH_2_: prostaglandin H_2_.

**Figure 3 life-14-01037-f003:**
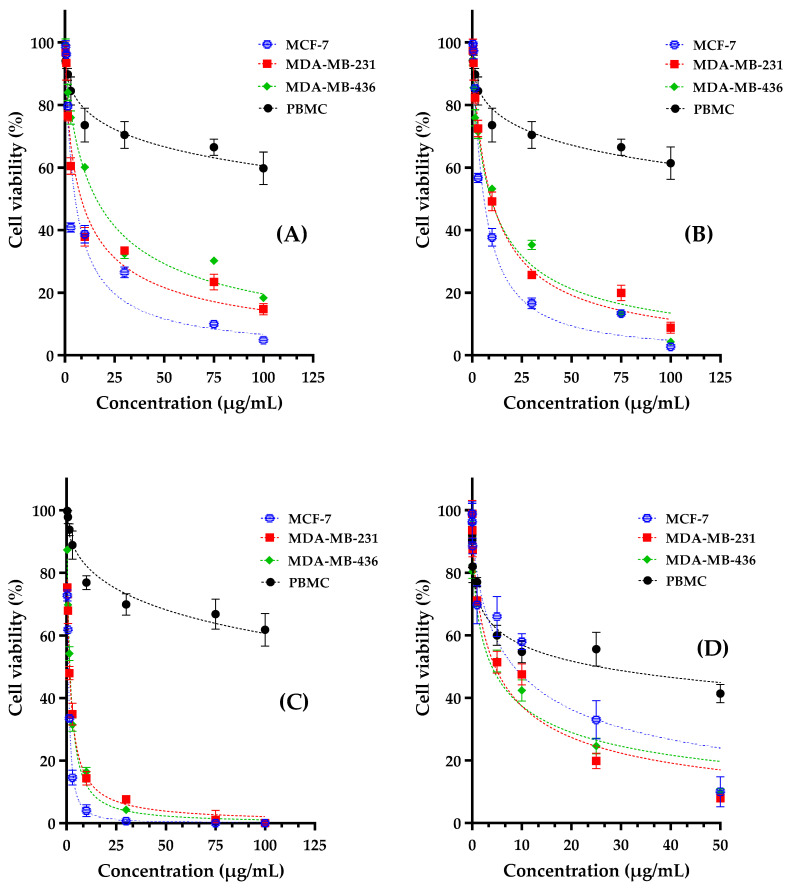
Cell viability of MCF-7, MDA-MB-231, MDA-MB-436, and PBMC after 72 h of treatment with thymol (**A**), carvacrol (**B**), and thymol/carvacrol combination (**C**) and cisplatin (positive control, (**D**)) using MTT test.

**Table 1 life-14-01037-t001:** Physico-chemical properties, drug-likeness, and the bioavailability of thymol and carvacrol.

Characteristics	Thymol	Carvacrol
**Chemical structure**	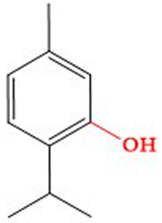	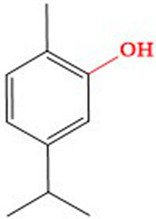
**Physico-chemical properties and drug-likeness**		
MW (g/mol)	150.221	150.221
TPSA (Å²)	20.23	20.23
Num. H-Bond acceptors	1	1
Num. H-Bond donors	1	1
Rotatable bonds	1	1
LogP	2.82	2.82
Lipinski *	Yes, No violation	Yes, No violation
Egan **	Yes	Yes
Veber ***	Yes	Yes
Bioavailability score	0.55	0.55
Bioavailability radars	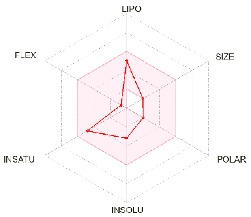	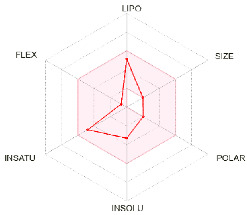

MW: molecular weight; TPSA: topological polar surface area; LogP: octanol–water partition coefficient; * Lipinski’s rule of five: MW ≤ 500; HBD ≤ 5; HBA ≤ 10; LogP ≤ 5. ** Egan rule: 0 ≤ LogP ≤ 5.88; TPSA ≤ 131 Å². *** Veber rule: rotatable bonds ≤ 10; TPSA ≤ 140 Å².

**Table 2 life-14-01037-t002:** In silico pharmacokinetic characteristics and toxicity of thymol and carvacrol.

In silico Prediction	Thymol	Carvacrol
	ADME Prediction Absorption-Related Parameters	
**Water Solubility**	−3.01	−3.31
Solubility Class	Soluble	Soluble
Caco-2 Permeability	−4.44	−4.41
Intestinal Absorption	Absorbed	Absorbed
Log K*_p_* (cm/s)	−2.64	−2.64
	**Distribution-related parameters**	
VDss	2.91	2.72
BBB Permeability	−1.82	−1.99
	**Metabolism-related parameters**	
CYP2D6, and CYP3A4 Substrate	No	No
CYP2D6, and CYP3A4 Inhibitors	No	No
	**Excretion-related parameters**	
Total Clearance	8.67	8.65
Renal OCT2 Substrate	Non-inhibitor	Non-inhibitor
	**Toxicity-related parameters**	
Predicted LD50 (mg/kg) Predicted Toxicity Class	640 4	810 4
Hepatotoxicity Probability	Inactive 0.75	Inactive 0.75
Nephrotoxicity Probability	Inactive 0.72	Inactive 0.72
Carcinogenicity Probability	Inactive 0.60	Inactive 0.60
Immunotoxicity Probability	Inactive 0.93	Inactive 0.96
Mutagenicity Probability	Inactive 0.99	Inactive 0.99
Cytotoxicity Probability	Inactive 0.89	Inactive 0.89

Water solubility: log mol/L, M.S.: moderate solubility; VDss: log L/kg; total clearance: log (mL/min/kg).

**Table 3 life-14-01037-t003:** Antioxidant activity of thymol, carvacrol, and their equal combination.

EO/Reference	DPPH Scavenging Capacity IC_50_ (µg/mL)	ABTS Scavenging Activity IC_50_ (µg/mL)	Oxygen Radical Absorbance Capacity (mmol TE/g)
Thymol	161.02 ± 6.89	125.31 ± 6.25	2.341 ± 0.009
Carvacrol	249.09 ± 9.04	107.88 ± 4.46	3.535 ± 0.127
Thymol/Carvacrol (1:1)	43.82 ± 2.41	23.29 ± 0.71	0.273 ± 0.096
Ascorbic acid (AA)	188.24 ± 6.46	25.20 ± 3.06	-

**Table 4 life-14-01037-t004:** IC_50_ values and selectivity indexes of thymol, carvacrol, thymol/carvacrol combination, and cisplatin on cancer cell lines (MCF-7, MDA-MB-231, and MDA-MB-436).

Treatments	IC_50_ Value ± SD (µg/mL) *				Selectivity Index **		
	MCF-7	MDA-MB-231	MDA-MB-436	PBMC	MCF-7	MDA-MB-231	MDA-MB-436
Thymol	5.16 ± 0.71	8.22 ± 0.94	15.77 ± 1.69	307.3 ± 3.48	59.55	37.38	19.44
Carvacrol	5.93 ± 0.77	9.76 ± 0.23	9.50 ± 1.15	343.0 ± 2.53	57.84	35.03	36.10
Thymol/Carvacrol (1:1)	0.92 ± 0.09	1.46 ± 0.16	1.70 ± 0.22	246.3 ± 3.78	267.71	168.96	144.88
Cisplatin	8.43 ± 0.92	4.60 ± 0.66	3.96 ± 0.59	25.36 ± 1.40	3.01	5.51	6.40

* Values are obtained from three independent experiments and expressed as means ± SD. ** Selectivity index = (IC50 of PBMC/IC50 of tumor cells).

## Data Availability

Data are available in the present paper.

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
