# Peer review of "Evaluation of the Interaction between Carvacrol and Thymol, Major Compounds of Ptychotis verticillata Essential Oil: Antioxidant, Anti-Inflammatory and Anticancer Activities against Breast Cancer Lines"

_life, 2024, doi:10.3390/life14081037_

Round 1

Reviewer 1 Report

Comments and Suggestions for Authors

·       More information about the chosen compounds, carvacrol and thymol from the literature should be cited in the introduction section.

·       The reason for the selection of these compounds should be emphasized.

·       Chemical structures of the compounds provided in Table 2 need clarity.

·       The heading number 3 is missing in the manuscript.

·       The probable mechanism of action for the reported activities may be added.

·       The results of the molecular docking study are too weak and look too preliminary.

·       For computational studies, specify the software versions, parameter settings, and models used. This detail is crucial for reproducibility and scientific rigor.

·       The results of the MD simulation are useless without further analysis. For how long the MD simulation was carried out? Binding free energy from MD simulation should be compared with those from docking. The compounds should be tested using ADME at least for the Lipinski rule-of-five.

·       According to the data given in Table 1, the DPPH radical scavenging potential was very high for Thymol/Carvacrol (1:1) (43.82). However, it was too low for the compounds when they are used separately as thymol with 161.02 and carvacrol with 249.09. I would suggest the authors justify this drastic variation and there was no correlation between antioxidant and anti-inflammatory activity results. The same kind of variation was observed in the inflammatory inhibition assay too.

·       Conclusion summarizes the main findings of the study. However, the authors may discuss briefly some potential applications for industry and consumers, as well as future perspectives on this research topic.

·       Mention the concentrations used for the standard drugs in the in vitro assays.

Author Response

Dear Editors and Reviewers,

We would like to express our gratitude for providing us with the opportunity to enhance our manuscript through the revised version, and we sincerely appreciate your valuable feedback. We are particularly thankful for your thorough review, which has significantly contributed to improving the manuscript's quality.

In response to the requested changes, we have carefully considered each recommendation by incorporating the suggested modifications or providing detailed responses. The manuscript has been adjusted in accordance with the author guidelines, and we have meticulously reviewed the author instructions to ensure compliance.

We assure you that all linguistic issues and typographical errors have been corrected in the manuscript. Once again, we sincerely thank you for the time devoted to our work, the efforts you have exerted, and the constructive feedback you have provided.

We hope that our revised manuscript will successfully address all your comments and meet the necessary criteria for publication in the Life journal.

Thank you, and please accept, Madam, Sir, the expression of our distinguished regards.

Reviewer 1:

 Comment 1.  More information about the chosen compounds, carvacrol and thymol from the literature should be cited in the introduction section.

Response 1. Thank you for your pertinent comment. Information on carvacrol and thymol has been quoted in the introduction.

Comment2. The reason for the selection of these compounds should be emphasized.

Response 2. Thank you. The reason why these compounds have been selected has been added to the introduction section.

Comment 3. Chemical structures of the compounds provided in Table 2 need clarity.

Response 3. We have provided clear chemical structures. Thank you for your remark.

Comment 4. The heading number 3 is missing in the manuscript.

Response 4. Thank you. The modification has been made.

Comment 5. The probable mechanism of action for the reported activities may be added.

Response 5. Thank you. The probable mechanism of action of the activities has been added to the discussion.

Comment 6. The results of the molecular docking study are too weak and look too preliminary.

Response 6. We thank the reviewer for this remark. We would like to clarify that the focus of our study was on the experimental evaluation of the antioxidant, anti-inflammatory, and anticancer activities of thymol, carvacrol, and their equimolar mixture, rather than on computational methods like molecular docking or simulations. The results presented are based on experimental data obtained through in vitro assays. We acknowledge that molecular docking studies could provide additional insights, but they were beyond the scope of this work. The primary aim was to establish the synergistic effects of these compounds through experimental validation. We hope this explanation addresses the concerns raised.

Comment 7. For computational studies, specify the software versions, parameter settings, and models used. This detail is crucial for reproducibility and scientific rigor.

Response 7. We thank the reviewer for this important observation. In our study, the computational analysis was conducted using bioinformatics websites rather than standalone software programs. Specifically, we used the SwissADME and PKCSM websites to evaluate pharmacokinetic parameters and the Pro-Tox III website to predict toxicity. These tools are well-recognized in the scientific community for their reliability and accessibility, allowing researchers to perform these analyses without the need for specialized software installations. Since these are web-based tools, they do not require specific software versions, parameter settings, or model installations, which is why we did not include such details in the manuscript. The results are reproducible by simply accessing these publicly available platforms and inputting the molecular structures as described. We hope this clarification aligns with the expectations for transparency and reproducibility in our computational analysis.

Comment 8. The results of the MD simulation are useless without further analysis. For how long the MD simulation was carried out? Binding free energy from MD simulation should be compared with those from docking. The compounds should be tested using ADME at least for the Lipinski rule-of-five.

Response 8. We thank the reviewer for this remark. We would like to clarify once again that our study did not involve any molecular docking or molecular dynamics (MD) simulations. Therefore, no MD simulation results were presented or analyzed in our manuscript. However, we did conduct a thorough ADME analysis, including the evaluation of drug-likeness according to Lipinski's rule of five, as well as other pharmacokinetic parameters. These results are already detailed in the results section of our manuscript. The focus of our study was to provide an in-depth experimental and computational evaluation of the pharmacokinetic properties of the compounds, without involving docking or MD simulations. We hope this clarification resolves any confusion and appreciate the opportunity to address these points.

Comment 9. According to the data given in Table 1, the DPPH radical scavenging potential was very high for Thymol/Carvacrol (1:1) (43.82). However, it was too low for the compounds when they are used separately as thymol with 161.02 and carvacrol with 249.09. I would suggest the authors justify this drastic variation and there was no correlation between antioxidant and anti-inflammatory activity results. The same kind of variation was observed in the inflammatory inhibition assay too.

Response 9. Thank you for your pertinent comment. We have added lines in the manuscript to justify the drastic variation observed and the fact that there is no direct correlation between the antioxidant and anti-inflammatory activity results.

Comment 10. Conclusion summarizes the main findings of the study. However, the authors may discuss briefly some potential applications for industry and consumers, as well as future perspectives on this research topic.

Response 10. Thank you. The conclusion has been modified following your recommendation.

Comment 11. Mention the concentrations used for the standard drugs in the in vitro assays.

Response 11. Thank you for this pertinent remark. We have added the concentrations used for the standard drugs used in our in vitro assays as recommended.

Reviewer 2 Report

Comments and Suggestions for Authors

1. Instead of guessing the probable route or cause of the synergistic effect, the author should perform bio-informatic studies to get the answer or explanation of the effect/ to get or know the interaction of different compounds with proteins in different ways.

2. Page 7, line 267, changes the words " different anti-inflammatory effects " as it is creating confusion. 3. Sometimes, it is irrational to rationalize the effect of two compounds together by using the single compound as a standard. The author should explain it. 4. Please explain or elaborate about "equimolar as a natural option."

Author Response

Dear Editors and Reviewers,

We would like to express our gratitude for providing us with the opportunity to enhance our manuscript through the revised version, and we sincerely appreciate your valuable feedback. We are particularly thankful for your thorough review, which has significantly contributed to improving the manuscript's quality.

In response to the requested changes, we have carefully considered each recommendation by incorporating the suggested modifications or providing detailed responses. The manuscript has been adjusted in accordance with the author guidelines, and we have meticulously reviewed the author instructions to ensure compliance.

We assure you that all linguistic issues and typographical errors have been corrected in the manuscript. Once again, we sincerely thank you for the time devoted to our work, the efforts you have exerted, and the constructive feedback you have provided.

We hope that our revised manuscript will successfully address all your comments and meet the necessary criteria for publication in the Life journal.

Thank you, and please accept, Madam, Sir, the expression of our distinguished regards.

Reviewer 2:

Comment 1.  Instead of guessing the probable route or cause of the synergistic effect, the author should perform bio-informatic studies to get the answer or explanation of the effect/ to get or know the interaction of different compounds with proteins in different ways.

Response 1. We thank the reviewer for this insightful suggestion. However, performing bioinformatics studies to explore the interaction of different compounds with proteins or to explain the synergistic effect was beyond the scope of our current study. Our research focused on the experimental evaluation of the antioxidant, anti-inflammatory, and anticancer activities of thymol, carvacrol, and their equimolar mixture. The results and discussions are based on experimental data obtained from in vitro assays. While we recognize the value of bioinformatics approaches, incorporating such studies would extend beyond our current research objectives. We focused on providing a thorough experimental analysis to support our findings, which are detailed in the results section of the manuscript.

Comment 2. Page 7, line 267, changes the words " different anti-inflammatory effects " as it is creating confusion.

Response 2. Thank you. The modification has been made.

 Comment 3. Sometimes, it is irrational to rationalize the effect of two compounds together by using the single compound as a standard. The author should explain it.

Response 3. We appreciate the reviewer’s comment and the opportunity to clarify our approach. In our study, the rationale for comparing the combined effect of thymol and carvacrol with their individual effects was based on the objective of understanding their potential synergistic interactions. While it is true that the activity of a mixture cannot always be predicted by the effects of its individual components, using single compounds as a reference standard is a common and established method in combination studies. This approach allows us to evaluate whether the combination exhibits enhanced, reduced, or comparable activity relative to the individual compounds. In our case, the equimolar mixture of thymol and carvacrol demonstrated a significant synergistic effect in several assays, which was substantially greater than the sum of their individual activities. This finding is particularly relevant in the context of developing natural therapies, as it suggests that the combination could provide a more potent therapeutic option than either compound alone. The comparison with single compounds helps to highlight this synergistic effect, providing a clearer understanding of the potential benefits of combining these compounds.

 Comment 4. Please explain or elaborate about "equimolar as a natural option.

Response 4. Thank you for your pertinent commentary. The term "equimolar as a natural option" has been well developed in the manuscript.

The use of an equimolar (1:1) blend of thymol and carvacrol is of particular interest as a natural option. This proportion enables their synergy to be exploited to the full, demonstrating exceptional results for antioxidant, anti-inflammatory and anti-cancer activities. The blend takes advantage of molecular interactions, significantly enhancing efficacy. This "whole-plant" approach preserves natural synergies and is often preferred [1]. What's more, the use of this natural blend offers the advantage of being a sustainable and ecological option, responding to the concerns of consumers and industries alike [2]. Thus, the development of products based on this blend has great potential as a natural alternative for the treatment of pathologies linked to oxidative stress, inflammation and cancer.

REFERENCES

  1. Bunse, M.; Daniels, R.; Gründemann, C.; Heilmann, J.; Kammerer, D.R.; Keusgen, M.; Lindequist, U.; Melzig, M.F.; Morlock, G.E.; Schulz, H. Essential Oils as Multicomponent Mixtures and Their Potential for Human Health and Well-Being. Front. Pharmacol. 2022, 13, 956541.
  2. Esty, D.C.; Winston, A. Green to Gold: How Smart Companies Use Environmental Strategy to Innovate, Create Value, and Build Competitive Advantage; John Wiley & Sons, 2009; ISBN 0470393742.

Round 2

Reviewer 1 Report

Comments and Suggestions for Authors

I wish to express my sincere appreciation to the authors for their revised manuscript with substantial revision. Their efforts have improved the manuscript content, significantly elevating the quality and clarity of the manuscript.